

# Estimating Soil Erosion Risk and Evaluating Erosion Control Measures for Soil Conservation Planning at Koga Watershed, Highlands of Ethiopia

Tegegne Molla[1,2] and Biniam Sisheber[1,2]

[1]Department of Geography and Environmental Studies, Bahir Dar University, Ethiopia
[2]Geospatial Data and Technology research Center (GDTC), Bahir Dar University, Ethiopia
*Correspondence to:* Tegegne Molla (tegmolla@gmail.com)

**Abstract.** Soil erosion is one of the major factor affecting sustainability of agricultural production in Ethiopia. This research applied recently validated RUSLE soil erosion model based on 6 to 14 years data for the Ethiopian highlands to estimate soil erosion and to evaluate soil conservation strategies in the Koga watershed. In this study, the average annual soil loss rate determined is 30.2 t ha$^{-1}$ yr$^{-1}$. The lowest soil loss is estimated about 12.1 t ha$^{-1}$ yr$^{-1}$ around the outlet of the Koga

river which is greater than the minimum tolerable soil loss (2 t ha$^{-1}$ yr$^{-1}$). The highest soil loss is estimated from the steep slopes of upper watershed which is 456.2 t ha$^{-1}$ yr$^{-1}$ by far higher than the maximum tolerable soil loss (18 t ha$^{-1}$ yr$^{-1}$). Using ordinary least square regression, existing land use aggravates soil erosion with 92% overall coefficient of determination ($R^2$). The topographic factor also influences soil loss with 89% $R^2$ value. Based on the evaluation of technical standards of soil conservation measures, 35.56% of the existing soil conservation practices fit with the recommended national standard

which has less contribution to tackle soil erosion. Therefore, to sustain agricultural practices, appropriate and standardized soil conservation structures on different erosion prone land uses and landforms should be implemented in Koga watershed.

## 1 Introduction

Livelihood of human kind is closely linked to soil and soil contributes food, clean water, clean air, and are a major carrier for biodiversity (Katsuyuki, 2009; Keesstra et al., 2016). Most of the people in the world remain heavily dependent on soil

resources as their main livelihood source that lead to high soil erosion. The high erosion rates are affecting mainly the developing countries due to intensive cultivation, deforestation, ploughing of marginal lands and extreme climate hazards (Biswas et al., 2015; Colazo and Buschiazzo, 2015; Ligonja and Shrestha, 2015). Soil erosion is worldwide environmental problem that threatens the lives of most smallholder farmers (Dai et al., 2015; Erkossa et al., 2015; Gessesse et al., 2015; Ochoa−Cueva et al., 2015; Taguas et al., 2015; Prosdocimi et al., 2016). Soil erosion rates beyond the tolerable limit changes in the hydro-

logical, biological, erosional and geochemical cycles, which result lack of the services that the soil offers to the human beings (Berendse et al., 2015; Brevik et al., 2015; Decock et al., 2015; Smith et al., 2015). On cultivated lands, appropriate soil conservation mechanisms supported with Vegetation are efficient strategies to control soil losses (Cerdà et al., 2016; Zhao et al., 2015). About 80% of the current agricultural land degradation is caused by soil erosion globally (Angima et al., 2003; Rodrigo



et al., 2015). Sustainable agricultural practice is challenged by severe soil erosion, as it reduces on-farm soil productivity and causes food insecurity (Sonneveld, 2003; Moges and Holden, 2006; Bewket, 2007). In most developing countries, including Ethiopia, anthropogenic activities trigger soil erosion (Belyaev et al., 2004; Hurni et al., 2005).

With the present Ethiopian population of 90 million with a growth rate of 2.7% (CSA, 2015), about 80% of the population
depends on agricultural practices leading to very high population pressure on the land. Together with corresponding high livestock density, the agricultural sector is leading to serious overuse of the land. As many literatures documented, soil erosion in highlands areas is seen as a direct result of the historical human settlement in Ethiopia because of its favorable climatic conditions, political factor and fertile soil (Hurni, 1993; Keesstra et al., 2016). Soil erosion in Ethiopian highlands is one of the biggest problems resulting in both on-site and off-site effects. This phenomenon has provoked by high population density,
overgrazing, deforestation, land fragmentation, steep terrains, and cultivation on marginal and fragile lands. Such factors aggravate soil erosion and productivity declines, resulting in food insecurity of small holder farmers. The annual rate of soil loss in the country is higher than the annual rate of soil formation rate (Hurni, 1993). Annually, Ethiopia losses over 1493 million tones of topsoil from the highlands due to erosion which can add about 1.5 million tons of grain to the country's harvest (Hurni, 1993; Lulseged and Paul, 2008; Yitbarek et al., 2012; Erkossa et al., 2015).

About 43% (537,000 km$^2$) of the total highland areas of Ethiopia are highly affected by soil erosion with an estimated average of 20 t ha$^{-1}$ yr$^{-1}$ and measured amounts of more than 300 t ha$^{-1}$ yr$^{-1}$ on specific plots (Hurni, 1990; Paulos, 2001; USAID CRSPT, 2000). Whereas the Blue Nile basin lost fertile soils with a rate of 131 million t yr$^{-1}$ soil (Betrie et al., 2011). Lack of appropriate soil conservation measures and poor land use management have played a great role for serious soil erosion problems in the country. So far, little or no sufficient documents have been available on the contribution of the different soil
conservation measures implemented on soil loss reduction at Koga watershed. The case of the study area (Koga watershed) as part of the highlands of Ethiopia and upper part of Blue Nile basin is affected soil erosion problems. Soil erosion risk differs spatially because of heterogeneous topography, geology, geomorphology, soil types, land cover, and land use. This research has estimated soil loss rates and evaluated soil conservation measures at Koga watershed based on an integrated approach of RUSLE soil erosion model and field measurements.

## 2   Material and Methods

### 2.1   Description of the study site

The study was carried out at Koga watershed which is one of the major river watersheds at the source of river Blue Nile, in North western Ethiopia (Fig. 1). It is located in the central highland ecoclimatic zone of Ethiopia between $11^o10^{'}06^{"}$ to $11^o24^{'}22^{"}$N latitude and $37^o02^{'}48^{"}$ to $37^o17^{'}41^{"}$E longitude surrounded by high mountains rise up to 3100m which serves as
the main source of water streaming in the rivers that feeds the Koga irrigation dam. Whereas the lowland part of the watershed is gently sloping which is 1880m above sea level.

In the upper catchment of the study area, more than 60% of the land is under intensive cultivation whereas more than 80% of the watershed area is under cultivation in the lower catchment below Koga dam. The area has Tepid Moist Mid Highland





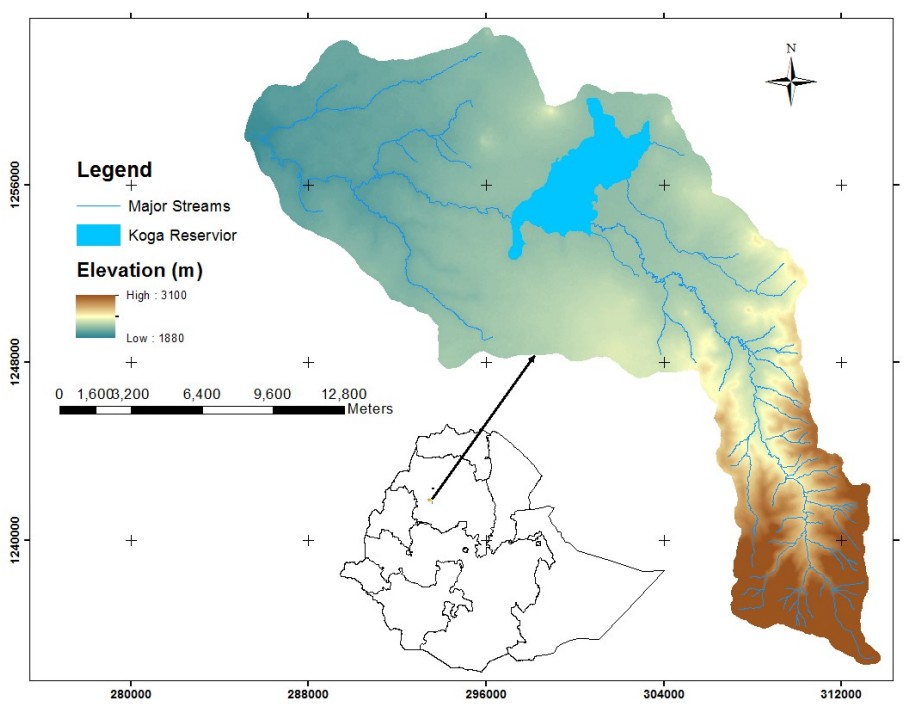

**Figure 1.** Location of Koga watershed

agro climatic zone affected by the position of north-south oscillation of Inter tropical Convergence Zone characterized by high annual rainfall variability. The monthly average (2000 to 2015) of Merawi station illustrates that 94% of falls occur in the months between May and October. Due to rugged topography and rainfall extremes, the probability of removing soil and aggravating mass movement is so high.

## 2.2 Research Methods

Soil loss at watershed level is caused by the interplay of physical, hydrological and land management practice. Therefore a mixed approach of field investigation and integrated RUSLE modelling are being adopted for soil erosion assessment for development of SWC measure scenarios. To assess the rate of soil loss in Koga basin and mapping erosion risk and conservation sites, topographic surface, meteorological, geological soil type, land use and SWC practices were collected (Fig. 2). Representative land use and land cover classes was generated from SPOT image (2.5×2.5m spatial resolution) supported with ground data. Evaluation of existing SWC measures have been evaluated at Asanat, Debreyakob and Rim sub-watersheds at upper, mid and lower positions of Koga catchment respectively.

Due to limitations in the availability of soil data, a combination of two different types of soil data were used in this study. The digital soil map produced by Ministry of water resource of Ethiopia using the FAO-UNESCO-ISRIC soil classification system and the Koga irrigation project pre-visibility study (ACRES, 2001) soil data were integrated to get a more accurate and





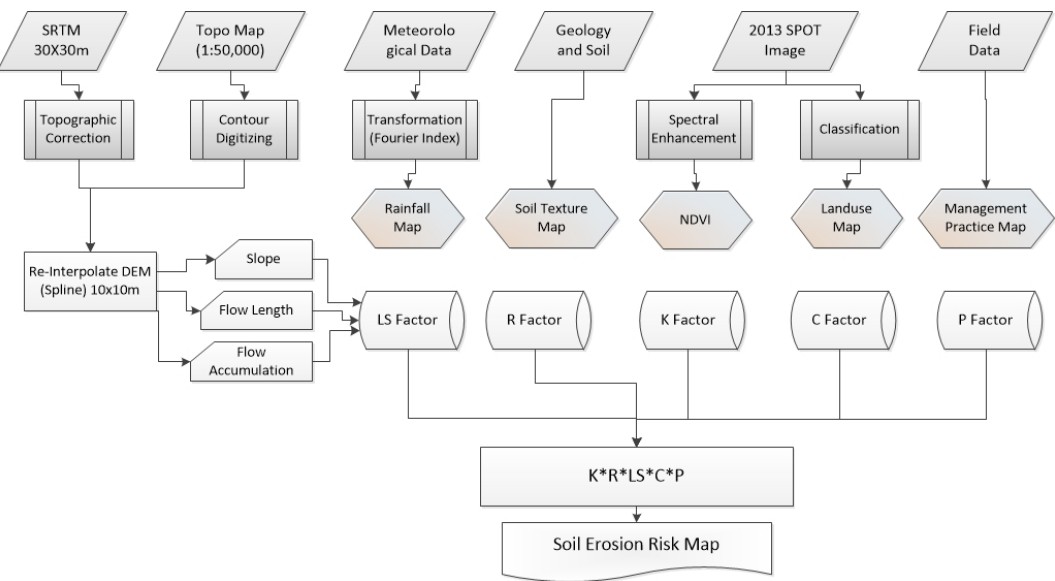

**Figure 2.** The conceptual framework of the research methodology

detailed soil map. Some of the soils of the lower catchment around and below the dam are poorly drained due to the formation of perched water table during the rainy seasons.

Devastation of existing soil conservation measures and poor quality of design significantly aggravate soil erosion processes in the study area. Major soil and water conservation structures were identified. Using simple random sampling method sample

5 plots were selected to evaluate the land use, farm bunds and check dam structures on the top, middle and lower part of the watershed. For evaluating existing land use, farm bund and check dam parameters a total of 27, 21 and 21 soil conservation structures respectively were measured. Treatment-Oriented capability classification scheme that has been tested in the northern highlands of Ethiopia and the national SWC guideline (MoARD, 2005) were applied for the present work.

## 10 3 Result and Discussion

Soil erosion risk differs spatially in the study area because of its rugged topography, geomorphology, landform, soil types, land cover, and land use. The rainfall erosivity, soil erodibility, slope length, slope steepness, cover-management and erosion control practice parametric factors were prepared and converted into same resolution raster layers. The RUSLE model is multiplication of unique value of each spatially corresponding grid cell in the six raster layers. Adaptation of the USLE to the

15 Ethiopian-Eritrean Highlands conditions was carried out by Hurni (1985) using data from two to five years of research in six Soil Conservation Research Projects (SCRP) stations (without Dizi, as it was not established at this time). For the estimation of soil loss, the following data analysis from different sources was used.



### 3.1  Rainfall Erosivity (R- Factor)

Hurni (1985) developed and used an empirical equation $(-8.12 + 0.562P)$ to estimates R$-$value for the Ethiopian highlands from annual total rainfall (P) from two to five years of rainfall data. But after Hurni, a value of 0.88 high correlation for monthly precipitation $(x)$ and monthly erosivity was found in the Ethiopian highlands using the regression equation of $(0.55x - 4.7)$

5    (Kaltenrieder, 2007) from 14 years of rainfall data and this equation is used in this research. Considering topographic variation, the R factor was determined from long term rainfall data collected from six different climatic stations: Meshenti, Adet, Merawi, Tissabay, Durbete, and Dangla from the year 2000 to 2015. There is spatial variation of rainfall caused by topographic variations and the downstream receives less rainfall than the plain of the upstream watershed (Fig. 3). The range of the rainfall erosivity computed varied from 1481 to 1885 MJ mm ha$^{-1}$ yr$^{-1}$ with the average value of 1404 MJ mm ha$^{-1}$ yr$^{-1}$. R-values are reliable for the study area with an average erosivity validated from SCRP experiments from the same agro ecological zone.

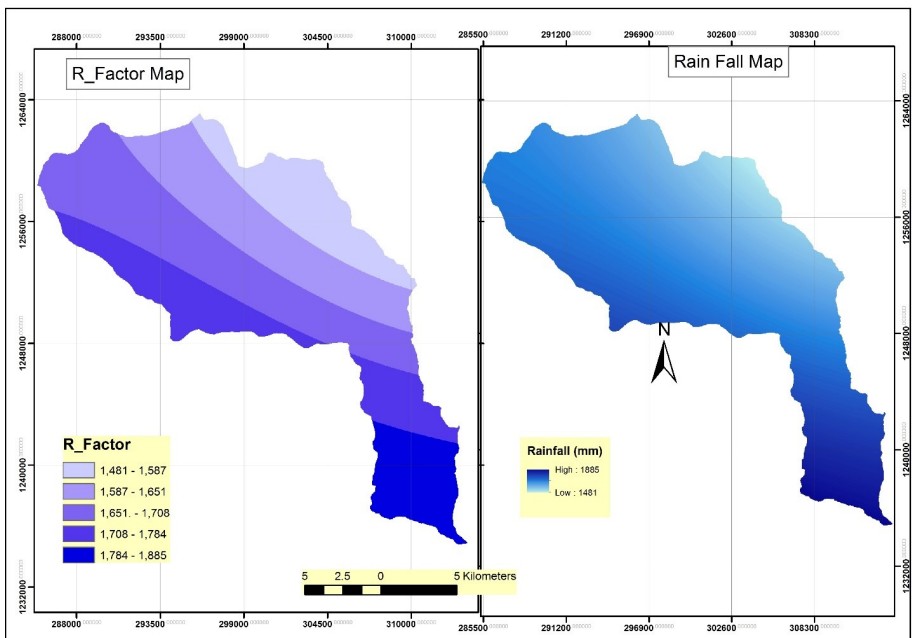

**Figure 3.** R factor map of the study area

### 3.2  Soil Erodibility Factor (K)

The soil erodibility factor reflects the combined effect of soil properties of texture, permeability and organic matter. For the area of Koga watershed, there are totally ten types of soil classes in the study area (table 1). The dominant soil type is Haplic Alisol.

15    K factor for each soil type were estimated by its relation with soil characteristics based on the detailed soil map (Fig. 4). Finally





**Table 1.** Soil types and their characteristics in Koga watershed

| Soil | Soil units | Soil type | characteristics | Area (ha) |
|---|---|---|---|---|
| Pd/v | Eutric Vertisols | Cracking heavy clay | Poorly to very poorly drained, very deep, very dark when dry, friable, cracking heavy clay | 1160.2 |
| Pd/g | Eutric Gleysols | Sandy clay loam to clay | Very poorly drained, very deep, friable, acid | 2824.5 |
| UpA | Haplic Alisols | Very friable to friable clay loam to clay | well drained, very deep, strongly acid | 10502.5 |
| Upb | Haplic Alisols | Very friable to friable Clay loam to clay | Same as in UpA, but with complex 2 to 5% slope | 5547.3 |
| Mr | Lithic Leptosols | Extremely rocky silty clay loam to silty clay | Excessively drained, very shallow soil | 87.2 |
| Upc | Haplic Alisols | Very friable to friable clay loam to clay | Same as in UpA, but with simple slopes of 5 to 15% | 1355.2 |
| Pf/t | Gleyic and Chromic Cambisols | Silty clay loam to silty clay | Moderately well to imperfectly drained, very deep, acid | 196.8 |
| Pd/gd | Eutric Gleysols | Sandy clay loam to clay | Same as in Pd/g, but with a better drainage during the dry periods due to proximity to incised Koga river | 784.4 |
| Pd/gb | Eutric Gleysols | Sandy clay loam to clay | Same as in Pd/g, but with complex 2 to 5% slope | 316.3 |
| Md | Luvic Phaeozems and Chromic Cambisols | Friable sandy clay loam to clay | well drained, moderately deep to very deep, in places stony to very stony | 6270.8 |

the thematic soil types were reclassified to K values by using various data and recent studies adapted to Ethiopian highlands (SCRP, 2002; Erdogan et al., 2007; Kaltenrieder, 2007; Andersson, 2010). The spatial distribution map of K factor derived from the final soil map (Fig. 4).

The erodibility map show that Lithic Leptosols and Eutric Gleysols are highly susceptible to soil erosion with K values of 0.32 and 0.31 respectively followed by the highland area Luvic Phaeozems and Chromic Cambisols that have 0.23 K value which are moderately sensitive to erosion.



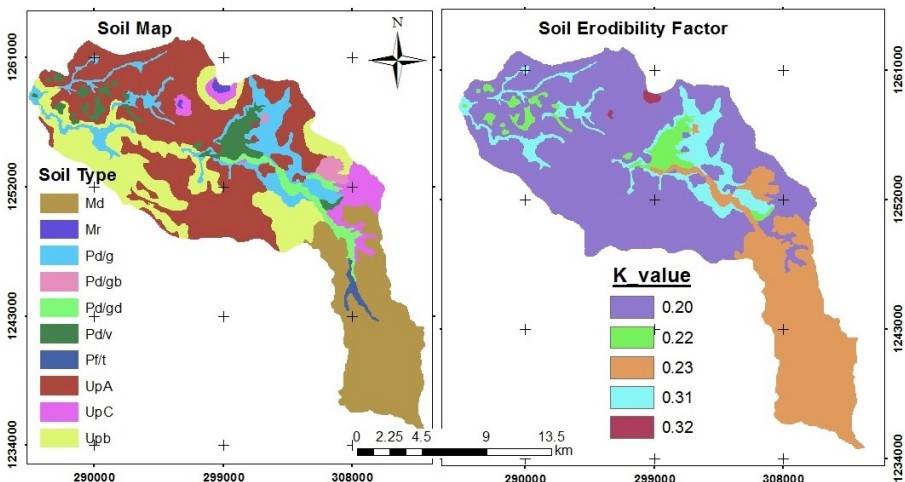

**Figure 4.** K factor map of Koga watershed

## 3.3 Slope length and steepness factor (LS)

The combined *LS* factor is the effect of slope length and slope steepness on soil loss. In RUSLE, the LS factor is relative to the site conditions with the "standard" slope steepness of 9% and slope length of 22 m plot (Wischmeier and Smith, 1978; Renard et al., 1997; Robert and Hilborn, 2000). Length and steepness of a slope affects the rate of soil erosion from study site. The slope length and steepness were computed from the DEM using ArcGIS Spatial analyst plus and arc hydro extension. Contours were digitized from a 1:50,000 topographic map prepared by Ethiopian mapping agency. DEM of the study area was interpolated using spline at 10m vertical interval and 0.01m vertical resolution from digitized contours.

As shown in figure 5, spatial variability of the LS factor has been derived from 20 meter resolution DEM of the study area based on the following equations (Renard et al., 1997; Kaltenrieder, 2007):

$$L = \left(\frac{\lambda}{22.31}\right)^m \tag{1}$$

Where

$$m = \frac{\beta}{(1+\beta)} \tag{2}$$

$$\beta = \frac{\left(\frac{\sin\theta}{0.0896}\right)}{\left[3\left(\sin\theta\right)^{0.8} + 0.56\right]} \tag{3}$$

$\lambda$ = the horizontal projection (meter)

$\theta$ = slope angle



The Steepness factor that derived from slope map of the study area is calculated for high slope (>9%) and low slope land (<9%) as shown below.

$S = 16.8 \sin\theta - 0.5$ (for slope angle $\theta \geq 9\%$)

$S = 10.8 \sin\theta + 0.3$ (for slope angle $\theta < 9\%$)

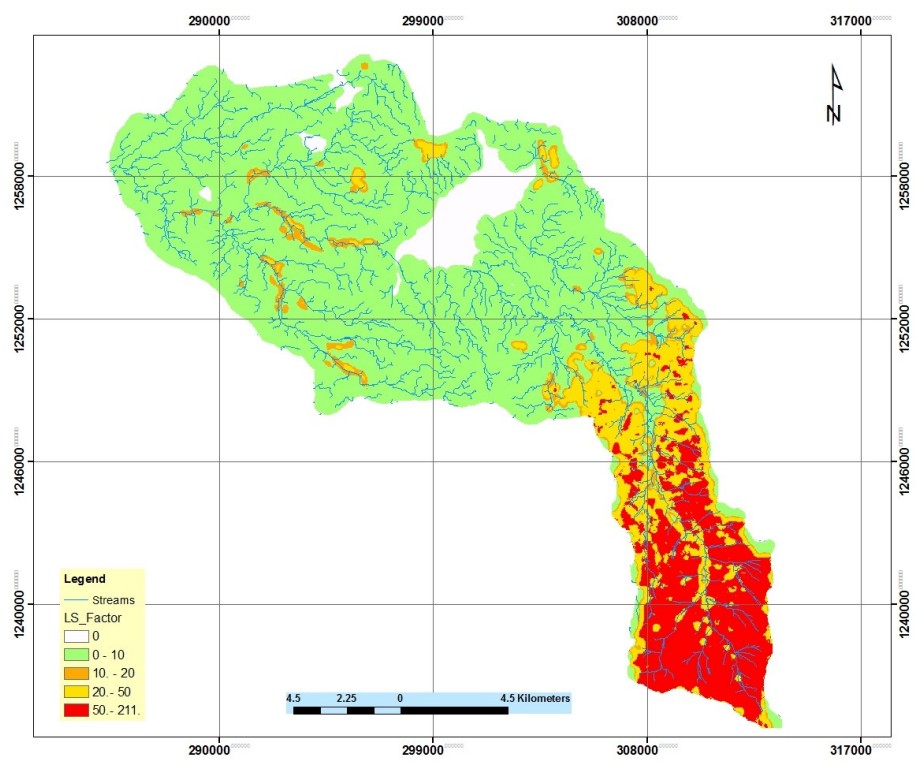

**Figure 5.** Slope length and steepness factor (LS) map of Koga watershed

The combined LS factor value in the study area varies from 0 to 211, with mean and standard deviation of 3.54 and 2.95 respectively. Majority of the lowland part of the watershed is gentle slope with minimum LS value.

### 3.4 Cover Management (C factor)

10   The C-factor represents the effect of plants, crop sequence and other soil cover surface on soil erosion. The cover management factor is the ratio of soil loss from vegetated plots versus bare soil plots with values between 0 and 1 (Wischmeier and Smith, 1978; Robert and Hilborn, 2000). In this study, C Factor is expressing qualitative properties of a specific plot which has to be artificially quantified using empirical process, i.e. expressed by a number, in order to be able to calculate soil loss using




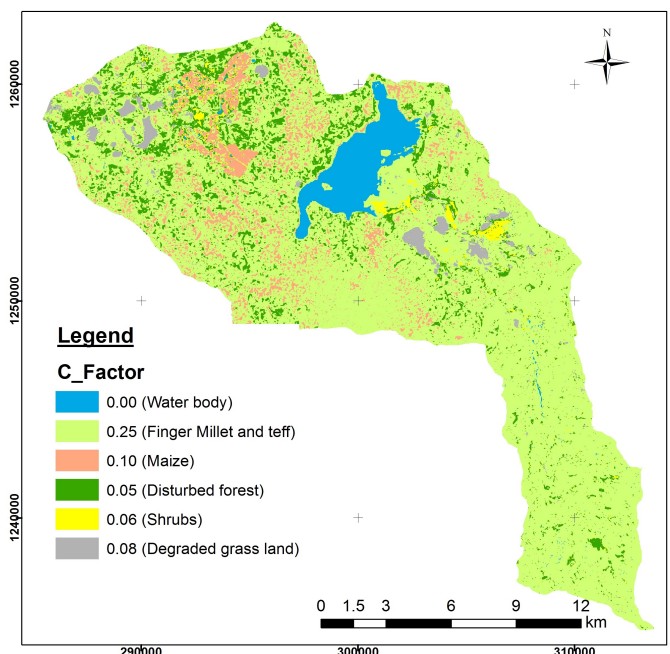

**Figure 6.** Map of land cover and their C values (after Hurni, 1985; Wischmeier and Smith, 1978)

RUSLE. Due to spatial and temporal variations, many studies used remote sensing data to classify land use and land cover units for quantification of C values with intensive ground truth (Beskow et al., 2009; Karydas et al., 2009; Tian et al., 2009). For the study area, representative land use classes were delineated. C-factor is determined from land cover classes as shown in Fig. 6. Land cover of the study area consists mainly of agricultural lands. The long term reliable crop cover data in the Soil Conservation Research Project (SCRP 2000*a-f*, 2002) database files and reports (SCRP 1982, 1983, 1984, 1986, 1988, 1991, 2000, 2002) were used to calculate the mean C values by averaging each record for a particular land use.

### 3.5    Erosion Management Practice (P-factor)

Erosion management practice is also one of the factor that has significant effect on soil erosion rate. The entire study area is not treated with permanent soil and water conservation measures; there is only a small area that has been treated with terracing, strip cropping, mulching and stone cover. Due to lack of data on permanent conservation measures, P-values suggested by Wischmeir and Smith (1978), Shi et al. (2002) and Bewket and Teferi (2009) are used (table 2).  The P-values are assigned by delineating the land in to 6 major land uses such as arable land, forest, grass land, shrub, water body and wetland. These management activities vary on the slopes of the agricultural land. The arable land is sub-divided in to six classes based on



**Table 2.** P value in different slope classes

| No | Land use type | Slope class | P factor |
|----|---------------|-------------|----------|
| 1 | Arable land | 0-2% | 0.10 |
| 2 | Arable land | 2-5% | 0.18 |
| 3 | Arable land | 5-8% | 0.24 |
| 4 | Arable land | 8-15% | 0.39 |
| 5 | Arable land | 15-30% | 0.45 |
| 6 | Arable land | >30% | 0.5 |
| 7 | Forest | all | 0.001 |
| 8 | Grass land | all | 0.02 |
| 9 | Shrub | all | 0.01 |
| 10 | Water body | all | 0.00 |
| 11 | Wetland | all | 0.00 |

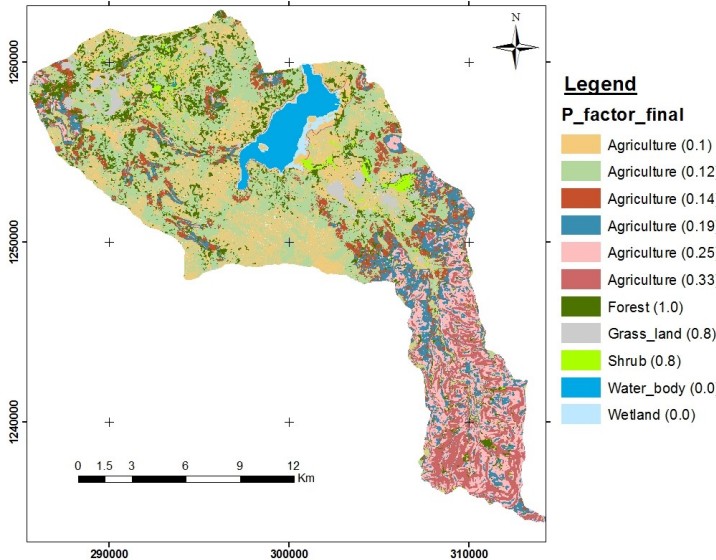

**Figure 7.** Spatial distribution of P factor map (Wischmeier and Smith, 1978; Shi et al., 2002; Bewket and Tefera, 2009)

the slope percent to assign different P-value for each slope classes (0-2%, 2-5%, 5-8%, 8-15%, 15-30%, and >30%). P value



ranges from 0 to 1 depending on the soil management practices employed on the farmers' plot of land (Fig. 7). Open areas with no conservation measures are given a value of 1.

### 3.6 Annual Soil Loss Estimation within Koga Watershed

The annual soil loss rate was determined by a cell-by-cell analysis of the soil loss surface by multiplying the respective RUSLE factor values computed by the Equation:

$$A = R * K * L * S * C * P \tag{4}$$

Where A is the annual soil loss (metric t ha$^{-1}$ yr$^{-1}$) resulting from sheet and rill erosion. On the fact that RUSLE does not account for gully erosion or mass movements. These parameters were determined based on model calibrated for Ethiopian Highlands by Kaltenrieder (2007). Various researchers and model initiators made a first attempt to adapt the USLE to the Ethiopian-Eritrean Highland conditions using 2 up to 5 years of measurements in six research stations available at that time (Hurni, 1985; Renard et al., 1997; Nyssen et al., 2008). This research used the USLE model parameters adapted and validated to the Ethiopian Highlands using 6 up to 14 years of measurement in seven research stations. Considering the agro climatic condition and data availability, spatial differences of soil erosion in the watershed have been investigated using RUSLE model and survey of soil erosion indicators. Based on FAO (1986) basic classification of desertification and Gebreyesus and Kirubel (2009), the categorization of different soil erosion classes (Fig. 8) were done with some modification to suit the features of the study area. The normal Soil loss tolerance (SLT) determined by Hurni (1985) to Ethiopia condition range from 2 to 18 t ha$^{-1}$ yr$^{-1}$.

On average, the rate of annual soil loss in the Koga watershed was predicted as 30.2 t ha$^{-1}$ in with specific spot exhibiting losses of 716 tones at the upper part of the watershed. The annual soil loss computed in Koga watershed is higher than the normal Soil loss tolerance. The total soil loss was modeled as 10.8 million t ha$^{-1}$ from 29,524 hectares of land. This research result has the same pattern with previous researches conducted on similar agroecological zones. For instance, FAO estimated 100 t ha$^{-1}$ yr$^{-1}$ soil loss from cropped lands in the highlands of Ethiopian in which Koga watershed is included. Soil Conservation Research Program (SCRP) also conducted a study at Anjeni research station showed that the annual soil loss rate to be 131 - 170 t ha$^{-1}$ (SCRP, 1996; Betrie et al., 2011). Figure 8 shows the spatial distribution of annual soil loss estimates using RUSLE model that ranges from 12.1 t ha$^{-1}$ at the outlet to 456.2 t ha$^{-1}$ at the upper part of the study area for 2015. High soil loss rates were observed at the upper parts of the study area and along the sides of rivers. Soil loss rates from these areas were above 50 t ha$^{-1}$ yr$^{-1}$ (Fig. 8). The statistical values in table 3 revealed that the rate of soil loss has a significant correlation with the slope conditions in the area.





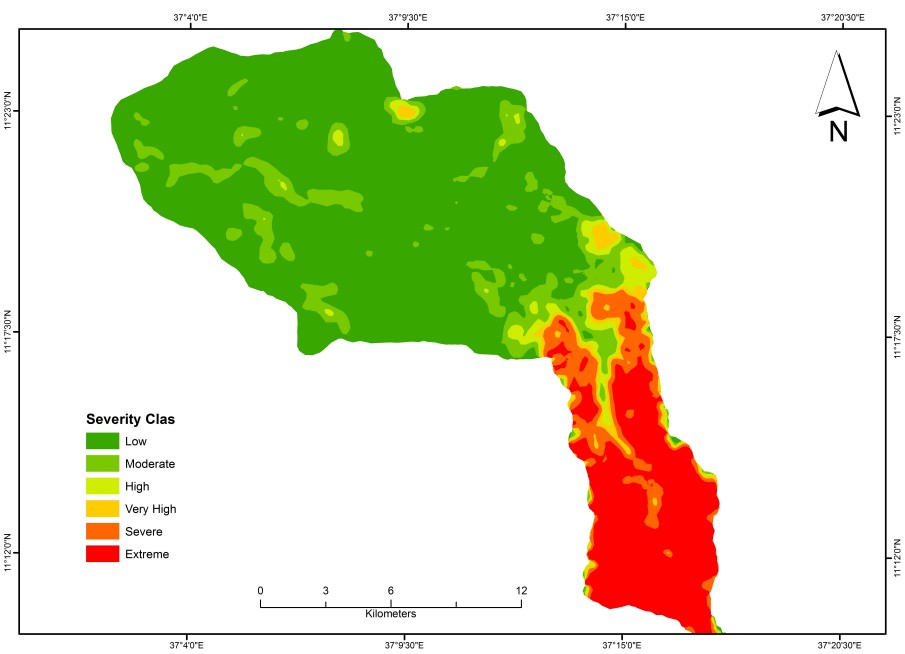

**Figure 8.** spatial distribution of soil loss estimates

**Table 3.** Annual soil loss severity classes

| Soil loss (t ha$^{-1}$ yr$^{-1}$) | Severity class | Area (ha) | Percent of total area | Total Annual soil loss (ton) | Percent of total soil loss | Average Slope (%) |
|---|---|---|---|---|---|---|
| 0-5 | Low | 19,341.2 | 62 | 671,849 | 6 | 3 |
| 5-20 | Moderate | 3,430.7 | 11 | 332,427 | 3 | 6 |
| 20-50 | High | 939.8 | 3 | 198,645 | 2 | 10 |
| 50-100 | Very High | 651.2 | 2 | 298,361 | 3 | 12 |
| 100-150 | Severe | 1,706.9 | 6 | 1,286,745 | 12 | 17 |
| 150-716 | Extreme | 4,924.3 | 16 | 8,077,658 | 74 | 26 |
| Total | | | | 10,865,685 | 100 | |



## 3.7 Driving forces of erosion

Ordinary least square regression analysis on 13077 hill slopes locations of the entire watershed indicates soil loss has high correlation with land use/ land cover and topographic factors. The overall coefficient of determination ($R^2$) is 92% for the former and 89% for the later. The result indicates the influence of land use and topography for soil loss in those randomly sampled hill slope locations show strong correlation in the entire basin. 74% of the erosion comes from steep slope part of the watershed. GIS zonal analysis of the effect of land use and cover on soil loss per land use is compared (Table 4). Cultivated land show the highest annual soil loss because of cultivation of steep slopes. 80% of the total soil loss comes from cultivated land followed by grass land. From field observation it was also found that there is cultivation on the steep slope side of the mountains and hills. Uncontrolled free grazing in the communally owned grass land is a common practice triggering high soil loss.

In the foot slope part of the hill slope on the other hand, high erosion occurs at the banks of the major streams and the

**Table 4.** Total annual soil loss on different land uses

| Land use type | Area (ha) | Area (%) | Total Annual soil loss (tons) | Annual soil loss (%) |
|---|---|---|---|---|
| Water body | 1700 | 6 | - | - |
| Wetland | 400 | 1 | 25585 | 0.3 |
| Forest cover | 300 | 1 | 784790 | 7.2 |
| Shrub land | 3100 | 10 | 52025 | 0.5 |
| Grass land | 7500 | 25 | 1317264 | 12.1 |
| Cultivated land | 16700 | 56 | 8686020 | 79.9 |
| Total | 29700 | 100 | 10865685 | 100 |

reservoir because of extensive cultivation and over grazing on friable soil where there is less vegetation cover. The effect of soil erodibility is also another factor for the variation of erosion in the basin. The soil types constituting 32% (table 1) of the area mostly in the upstream are characterized by poorly to moderately drained, stony and shallow soil type having moderate infiltration rates; altogether results high erodibility factor. Therefore high erosion in Koga Basin is strongly attributed to land use problem, free grassing and cultivation on steep slope hills and mountains especially in a longer slope length and convex shape.

## 3.8 Evaluation of SWC structures

The basis for the implementation of the SWC interventions on a large scale was the 1975 land reform. After analyzing the risks of soil, water and land degradation, the government of federal democratic republic of Ethiopia (FDRE) has intensively launched the natural resource development work through public mobilization since 2010 (Badege, 2001; MoARD, 2010). The



SWC activities were carried out using food aid in the form of food-for-work.

The RUSLE is one of the critical tools for the assessment of the situation concerning erosion in a specific area. The factors do help give information about the soil erosion symptoms. In addition to RUSLE, evaluation of soil conservation measures provides information about the backgrounds of erosion symptoms for designing appropriate solutions to the problem. This study evaluated the current land uses against treatment oriented capability classification (TOCC) prioritization (MoARD, 2005) for soil conservation planning. Existing soil conservation practices on 27 farm plots (10 plots on the middle and 17 plots at the upper watershed) with dimension of 100m*100m at every 200 meters spacing were identified and compared with the standardized practices which will form the basis for future monitoring of soil erosion and sedimentation at Koga dam and the watershed as a whole. Rating of existing land use type versus its required soil and water conservation structures is presented in table 5 based on the information collected from the field.

Table 5: Evaluation of current land use against TOCC scheme at upper and middle Koga watershed

| Plot no. | Soil depth (m) | Slope (%) | ESCP* | ReSCP** | Rating fitness of ESCP Vs ReSCP (%) |
|---|---|---|---|---|---|
| 1 | 48 | 26 | stone face soil bund and Vegetative barrier | Contour cultivation, strip cropping, vegetative barrier and terraces | 50 |
| 2 | 52 | 12 | Contour cultivation, damaged stone made terrace | Contour cultivation, strip cropping, vegetative barrier, broad-based terraces | 41 |
| 3 | 51 | 23 | No soil conservation structures | Bench terracing and terracing | 0 |
| 4 | 28 | 11 | Contour cultivation, stone face soil bund, vegetative barrier | contour cultivation, strip cropping, vegetative barrier, Broad-based terraces | 75 |
| 5 | 6 | 55 | No soil conservation structures | Tree plantation | 0 |
| 6 | 51 | 34 | No soil conservation structures | Bench terracing, hill side ditching, individual basins on less deep soils | 0 |
| 7 | 52 | 35 | No soil conservation structures | Terracing, hill side ditching, individual basins on less deep soils | 0 |





Table 5: continued ...

| | | | | | |
|---|---|---|---|---|---|
| 8 | 38 | 10 | Contour cultivation, stone made terrace | Contour cultivation, strip cropping, vegetative barrier, broad-based terraces | 50 |
| 9 | 76 | 10 | Contour cultivation, stone face soil bund, vegetative barrier | Contour cultivation, strip cropping, vegetative barrier, broad-based terraces | 75 |
| 10 | 51 | 16 | No soil conservation structures | Bench terracing or terracing | 0 |
| 11 | 56 | 17 | Contour cultivation, damaged stone terrace | Bench terracing, terracing | 70 |
| 12 | 71 | 9 | Contour cultivation, stone face soil bund | Contour cultivation, strip cropping, vegetative barrier, broad-based terraces | 50 |
| 13 | 45 | 11 | Damaged stone bund, contour cultivation | Contour cultivation, strip cropping, vegetative barrier, broad-based terraces | 45 |
| 14 | 51 | 18 | Contour cultivation, stone face soil bund, vegetative barrier | Bench terracing, terracing | 100 |
| 15 | 77 | 7 | No soil conservation structures | Contour cultivation, strip cropping, vegetative and rock barrier | 0 |
| 16 | 52 | 12 | Contour cultivation, damaged stone face soil bund | Bench terracing, terracing | 25 |
| 17 | 66 | 11 | Stone face soil bund, contour | cultivation Contour cultivation, strip cropping, vegetative barrier, broad-based terraces | 50 |



Table 5: continued ...

| | | | | | |
|---|---|---|---|---|---|
| 18 | 51 | 19 | Contour cultivation Terraces | Bench terracing, terracing | 25 |
| 19 | 55 | 18 | No soil conservation structures | Bench terracing | 0 |
| 20 | 130 | 4.5 | Stone face soil bund, contour cultivation | Contour cultivation, strip cropping, vegetative barrier, broad-based terraces | 50 |
| 21 | 55 | 13 | Damaged soil bund, contour cultivation | Bench terracing, terracing | 50 |
| 22 | 84 | 7 | Vegetative barrier, contour cultivation | Contour cultivation, strip cropping, vegetative barrier, broad-based terraces | 50 |
| 23 | 120 | 5.5 | Contour cultivation | Contour cultivation, strip cropping, vegetative and rock barrier, road-based terraces | 25 |
| 24 | 155 | 4 | No soil conservation measure | Contour cultivation, strip cropping, vegetative barrier, broad-based terraces | 0 |
| 25 | 134 | 4.5 | No soil conservation structures | Contour cultivation, strip cropping, vegetative barrier, broad-based terraces | 0 |
| 26 | 110 | 10 | Contour cultivation | Contour cultivation, strip cropping, vegetative barrier, broad-based terraces | 25 |
| 27 | 96 | 17 | damaged stone face soil bund | Bench terracing, terracing | 50 |
| Average | | | | | 33.56% |

*ESCP = Existing Soil Conservation Practice(s)

**ReSCP = Recommended Soil Conservation Practice(s)





On average 35.56% (standard deviation of 29.06%) of the existing implemented soil conservation practices fit with the national technical standards. Better matches with recommended one observed at the middle part of the watershed. The treatment oriented capability classification scheme prohibits the tillage on such types of very steep slopes and very shallow soil depth less than 10cm.

## 3.9 Evaluation of current and recommended soil conservation structures

From 63 farm bund structures surveyed, the horizontal distance and vertical intervals of only 22 spots (34.38%) were constructed based on the standardized package set on the national guideline (Table 6). The remaining 41 terraces do not met the minimum standards mentioned in the national standardized package. The horizontal distance between existing terraces range between 8 to 28 m, with an average of 14.7 m for slope 3% to 20% of plots. The main reason for the failure to achieve sustainable conservation structures is lack of knowledge and skill on construction practices. In addition, most farmers perceived that constructing bunds in narrow spacing may create difficulty in plowing activities. Large numbers of bunds reduce farm size at the same time needs much labor force to implement.

The result (Table 6) revealed that vertical interval (height of bund) of terraces is wider than the recommended value in which huge amount of runoff has been accumulated on the terraces. The average length of bunds at the watershed is 80 m with maximum of 122 meter and minimum of 15 m. from the survey result, 65.08% of the farm bunds have length above 80 meter which is out of the technical standard recommended by the guideline.

Gully prevention mechanisms, check dams constructed in both the upper and middle parts of the watershed failed due to poor foundation and lack of proper prone and spill ways. Large gullies are formed at the upper watershed in which poor integration of the physical and biological soil conservation measures. Based on the field check dam measurement, 4 stone check dams are constructed based on the standard, the rest check dams failed to meet the standard.

As depicted in figure 9, the average recommended spacing between check dams is 8.5 m, but actual average spacing measured is 9.42 m in which only 4 of check dams are fit the standard spacing. Most of the existing check dams failed due to incapability to overcome run-off from the up-slope areas of the watershed. Out of 16 check dam foundation on each survey plot, only 4 fulfill the technical standard set by the Ministry of Agriculture and 4 check dams missed foundations.

As a result, average height of the existing check dams at the watershed is less than half from the national standard. There is no gully stabilization with biological conservation measures done to minimize soil erosion and sedimentation. Inadequate and incorrect provision for safe outlets for excess runoff resulted for formation of fragile land around gullies. Poorly designed soil conservation structures, over grazing, deforestation and cultivating marginal land are main causes of soil erosion.



**Table 6.** Comparison of existing and recommended farm bund structures

| Plot No. | Slope (%) | Existing terraces | | | | | Recommended Dimension for Soil depth above 75cm | | |
| | | Numbers measured | VI (m) | HD (m) | Bund gradient | No of terraces VI dismantled | VI (m) | HD (m) | Bund gradient |
|---|---|---|---|---|---|---|---|---|---|
| 1 | 3 | 3 | 1 | 33 | 1 | 0 | 1 | 33 | 0.5-1 |
| 2 | 4 | 3 | 1.2 | 26 | 1 | 3 | 1 | 25 | 0.5-1 |
| 3 | 5 | 3 | 1.34 | 26 | 0.5 | 2 | 1 | 20 | 0.5-1 |
| 4 | 6 | 3 | 1 | 18 | 0.7 | 1 | 1 | 17 | 0.5-1 |
| 5 | 7 | 3 | 1.21 | 25 | 0.6 | 3 | 1 | 14 | 0.5-1 |
| 6 | 8 | 3 | 1 | 12 | 0.8 | 0 | 1 | 12 | 0.5-1 |
| 7 | 9 | 3 | 1.3 | 12 | 1 | 1 | 1 | 11 | 0.5-1 |
| 8 | 10 | 3 | 1.3 | 14 | 0.6 | 3 | 1 | 10 | 0.5-1 |
| 9 | 11 | 3 | 1.11 | 10 | 0.8 | 1 | 1.1 | 10 | 0.5-1 |
| 10 | 12 | 3 | 1.6 | 12 | 0.6 | 3 | 1.1 | 9 | 0.5-1 |
| 11 | 13 | 3 | 1.2 | 9 | 0.8 | 0 | 1.2 | 9 | 0.5-1 |
| 12 | 14 | 3 | 1.70 | 10 | 0.8 | 0 | 1.2 | 8 | 0.5-1 |
| 13 | 15 | 3 | 1.2 | 8 | 1 | 1 | 1.2 | 8 | 0.5-1 |
| 14 | 16 | 3 | 1.8 | 11 | 1 | 2 | 1.3 | 8 | 0.5-1 |
| 15 | 17 | 3 | 1.9 | 10 | 0.5 | 2 | 1.3 | 8 | 0.5-1 |
| 16 | 18 | 3 | 2 | 11 | 1 | 3 | 1.3 | 7 | 0.5-1 |
| 17 | 19 | 3 | 2 | 11 | 0.5 | 2 | 1.3 | 7 | 0.5-1 |
| 18 | 20 | 3 | 1.82 | 12 | 1 | 1 | 1.4 | 7 | 0.5-1 |
| 19 | 21 | 3 | 1.41 | 7 | 1 | 1 | 1.4 | 6 | 0.5-1 |
| 20 | 22 | 3 | 1.42 | 6 | 0.6 | 0 | 1.4 | 6 | 0.5-1 |
| 21 | 23 | 3 | 1.41 | 6 | 0.8 | 1 | 1.4 | 6 | 0.5-1 |
| Total | | 63 | | | | 30 | | | |



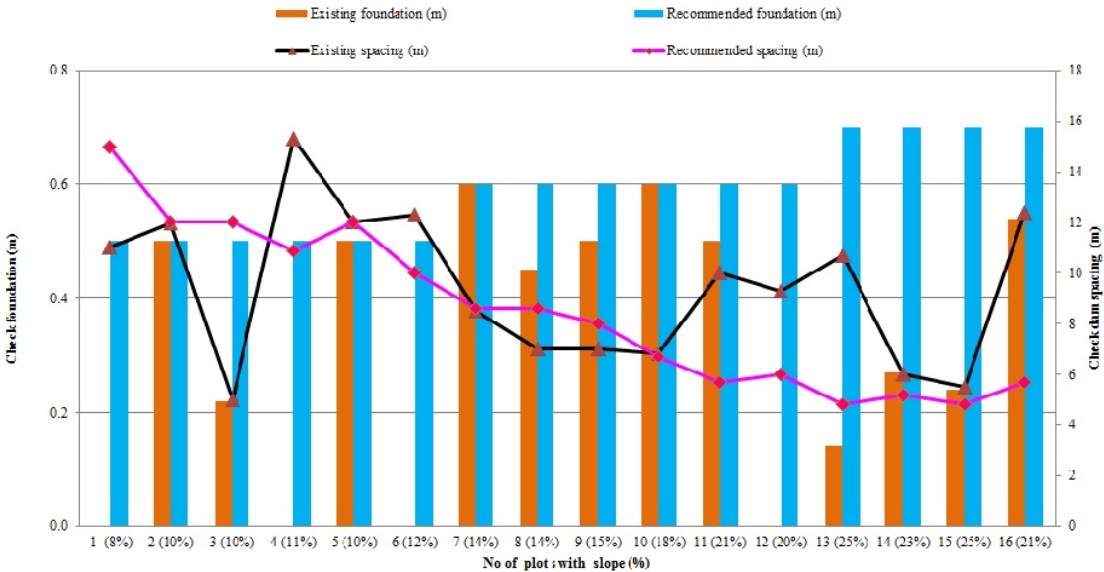

**Figure 9.** Existing and recommended check dam parameter

## 4   Conclusions

For this study, RUSLE was used to calculate soil loss rate which is adopted and validated for Ethiopian Highlands. The study reveals that remotely sensed data and GIS based approach becomes effective techniques to estimate watershed based soil loss rate in data scarce conditions. The RUSLE predict very high erosion with an average soil loss 30.2 t ha$^{-1}$ yr$^{-1}$ and total

5   soil loss is estimated as 10.8 million t ha$^{-1}$ yr$^{-1}$, which is very high. High erosion in the basin is caused by topographic factors shaping basin morphology; cultivation and over grazing on erosion sensitive locations such as on steep slope hills and mountains terrain units; banks of the river where the soil is fragile to be easily worn away.

Soil loss depends on the current land use and the type of soil and water conservation structures. In general, only 30.56% of different soil conservation practices fulfill the national technical standard of conservation measures at Koga watershed.

10   Even the constructed conservation structures are not managed properly. The common forms of erosion in the watershed are rill and sheet erosion coming from hillside, steep slope mountain and over cultivation. Therefore to minimize erosion risk in the watershed, standardized conservation measures considering local topographic variation have been constructed to sustain agricultural practice.

*Acknowledgements.* The authors gratefully thank for the financial support provided by Bahir Dar University, Ethiopia.



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
