# Peer review of "Estimating Soil Erosion Risk and Evaluating Erosion Control Measures for Soil Conservation Planning at Koga Watershed, Highlands of Ethiopia"

_Solid Earth, 2016_

## Referee Comment (RC1) · Anonymous Referee #1 · 14 Sep 2016

The paper examines soil erosion in the high lands and Ethiopia and evaluates the measures implemented to diminish it. The topic is interesting and very well presented in the Introduction. However, as I kept reviewing the paper I found that a major restructure needs to be done before being publishable in Solid Earth. I only check it until p. 8 since the number and magnitude of the changes to be done is so significant that it was time wasting to continue.

The main problem is related to the structure of the paper. The Results and Discussion section present mostly the way the data was obtained, but neither PRESENT nor DIS-

CUSS the data. Example: 3.3 you explain all the process to calculate it but then just 2 lines to discuss the results with no discussion. This section includes several parts (namely the beginning of subsections) which explain the methodology used in this paper. All referred to the methods implemented in this research should be moved to the Methodological section.

Then, the Results and Discussion should include the DESCRIPTION of the obtained results. What do your data show? What are the geographical patterns shown by your data? Present the values and infer patterns. After, you need to DISCUSS the factors explaining these data. What are the driving factors controlling soil erosion? You must refer to other studies that have dealt with similar topics. Are results similar/different) are the processes controlling soil erosion comparable and to what extent? Please, refer here to the existing literature on these topics.

Comments on specific sections p. 1, l. 13 "clean water and air" p. 1, l. 20 also geomorphic processes are affected p. 1, l. 22 "vegetation" p. 2, l. 9 "has been provoked" p. 2, l. 17 I don't understand this sentence, may be "Whereas" at the beginning is not appropriate p. 2, l. 27 space needed after elevation "m". Here and along the text. Usually the first time an altitude is mentioned you should add "above sea level" here you only mention it in the next sentence. p. 2, l. 31 rewrite this sentence, please. Maybe. . ."The lowland part of the watershed presents gentle slopes at elevations of 1880 m" p. 3, l. 3 high annual precipitations? Please give values p. 3, l. 12-17 This is methodology and should be moved there p. 4, l. 1-7 Idem p. 7, l. 2-20 Idem p. 8, l. 1-4 Idem Figure 2. The text is too small, not readable

---

## Referee Comment (RC2) · F. Pacheco (Referee) · 16 Sep 2016

REVISION Paper: se-2016-120 Title: Estimating Soil Erosion Risk and Evaluating Erosion Control Measures for Soil Conservation Planning at KogaWatershed, Highlands of Ethiopia

OUTLINE AND GENERAL APPRECIATION This is a conventional study on soil erosion rates estimated by the RUSLE equation, with indication of conservation measures for soil loss attenuation. It brings nothing conceptually new but it provides insights on soil erosion of a specific region of Ethiopia (the Koga watershed). The study is well written

and documented and merits publication in Solid Earth, with a minor revision.

CONCERNS I have no concerns on this paper

MINOR COMMENT 1) In lines 15-16 of page 1, the authors should include the environmental land use conflicts (ELUC) as another major cause of soil loss amplification, as recently recognized by Pacheco et al. (2014) and Valle Junior et al. (2014). The ELUC are related to land uses not conforming to soil's capability, meaning that are uses which deviate from the soil's natural use (e.g. practice of agriculture in soils solely capable of being used for forestry). Apart from the amplification of soil erosion, the ELUC have been demonstrated to provoke a decline in soil fertility (Valera et al., 2016). Somehow, these aspects of soil erosion / decline of soil fertility should be referred to in the revised manuscript.

RECOMMENDATION Minor revision 14 September 2016

REFERENCES Pacheco, F.A.L., Varandas, S.G.P., Sanches Fernandes, L.F., & Valle Junior, R.F. (2014). Soil losses in rural watersheds with environmental land use conflicts. Science of the Total Environment, v. 485–486C, p. 110–120.

Valle Junior, R.F., Varandas, S.G.P., Sanches Fernandes, L.F., & Pacheco, F.A.L. (2014). Environmental land use conflicts: A threat to soil conservation. Land Use Policy, v. 41, p. 172–185.

Valera, C.A., Valle Junior, R.F., Varandas, S.G.P., Sanches Fernandes, L.F., Pacheco, F.A.L. (2016). The role of environmental land use conflicts in soil fertility: A study on the Uberaba River basin, Brazil. Science of the Total Environment, v. 562, p. 463–473.

---

## Referee Comment (RC3) · Anonymous Referee #3 · 20 Sep 2016

This paper shows an interesting topic which is well presented in the Introduction. I would include in line 22 two new appointments:

Asensio, C.; Lozano, F.J.; Ortega, E.; Kikvidze, Z. Study on the effectiveness of an agricultural technique based on aeolian deposition, in a semiarid environment. Environmental Engineering and Management Journal, 14 (5): 1143-1150. 2015.

Lozano, F. J.; Soriano, M.; Martínez, S.; Asensio, C. The influence of blowing soil trapped by shrubs on fertility in Tabernas district (SE Spain). Land Degradation & Development, 24 (6): 575-581. 2013.

[Figure]

In the section on Material and Methods, I think part of the Description of the Study Site could be extended by reference to characteristics of the site that appear later in the Results and Discussion section. The part of Research Methods, should also be extended with a large number of descriptions in Results and Discussion. In the discussion could include more references to the effect of erosion, in that specific region of Ethiopia, on soil fertility.

I believe that this paper could be published in Solid Earth with a minor revision.

---

## Author Comment (AC1) · 20 Sep 2016

Authors' Response to the Referee #1 Comments Title of Paper: Estimating soil erosion risk and evaluating erosion control measures for soil conservation planning at Koga Watershed, Ethiopian Highlands Authors: Tegegne Molla and Biniam Sisheber We thank referee #1 for the comments and time spent to review this manuscript. The responses and explanations related to the comments are listed below:

Comment 1 (Paragraph 1): The paper examines soil erosion in the high lands and Ethiopia and evaluates the measures implemented to diminish it. The topic is interesting and very well presented in the Introduction. However, as I kept reviewing the paper I found that a major restructure needs to be done before being publishable in Solid Earth. I only check it until p. 8 since the number and magnitude of the changes to be done is so significant that it was time wasting to continue. Response: Thank you for your thorough review and salient observations. Since you didn't read the whole document, the comments lack convincing argument that leads to make significant changes. Besides, the nature of the research requires us to clearly discuses and present the RUSLE factors followed by integrated analysis of erosion and conservation which is presented from 3.6 to 3.9. Therefore, it is our sincere hope that this reply provides the necessary corrections and believed that the revised version can meet the journal publication requirements.

Comment 2: The main problem is related to the structure of the paper. The Results and Discussion section present mostly the way the data was obtained, but neither PRESENT nor DISCUSS the data. Example: 3.3 you explain all the process to calculate it but then just 2 lines to discuss the results with no discussion. This section includes several parts (namely the beginning of subsections) which explain the methodology used in this paper. All referred to the methods implemented in this research should be moved to the Methodological section. Response: Regarding the structure of the paper, there can be different approach. The methodology is a general framework of the study. However, the determination of RUSLE factors needs careful investigation and the formulas and values are different from other geographic areas. The factors explained boldly considering the agro climatic condition and data availability in the study area. That is why we presented the factors separately. Some discussions may be short, we did this to make this manuscript short, which is conceptually exploratory and we believe figures and maps are explanatory.

Comment 3: Then, the Results and Discussion should include the DESCRIPTION of the obtained results. What do your data show? What are the geographical patterns shown by your data? Present the values and infer patterns. After, you need to DIS-

CUSS the factors explaining these data. What are the driving factors controlling soil erosion? You must refer to other studies that have dealt with similar topics. Are results similar/different) are the processes controlling soil erosion comparable and to what extent? Please, refer here to the existing literature on these topics. Response: Under the Results and Discussion section Although it is very brief, we tried to show the result of our data and driving forces of erosion on 3.6 and 3.7. The maps clearly illustrates geographic pattern of soil erosion. In analyzing the driving forces we have reviewed different studies, for instance: • R- Factor (p. 5) is determined from long term data and cross validated from 14 years Soil Conservation Research Project experiments (SCRP is established by Centre for Development and Environment, Bern in Ethiopia). The result is reliable to use as input for RUSLE. • K- Factor (p. 5 & 6) is determined using recent studies conducted in Ethiopia (SCRP, 2002; Erdogan et al., 2007; Kaltenrieder, 2007; Andersson, 2010) and the authors discussed it shortly. • On subsection 3.3 (LS- Factor), we have added short discussions related to soil erosion as per the referee's suggestion. In general, the annual soil loss rate of Koga watershed is compared with other studies conducted within the same AEZ (subsection 3.6).

Comment 4: Comments on specific sections p. 1, l. 13 "clean water and air" p. 1, l. 20 also geomorphic processes are affected p. 1, l. 22 "vegetation" p. 2, l. 9 "has been provoked" p. 2, l. 17 I don't understand this sentence, may be "Whereas" at the beginning is not appropriate p. 2, l. 27 space needed after elevation "m". Here and along the text. Usually the first time an altitude is mentioned you should add "above sea level" here you only mention it in the next sentence. p. 2, l. 31 rewrite this sentence, please. Maybe: : :"The lowland part of the watershed presents gentle slopes at elevations of 1880 m" p. 3, l. 3 high annual precipitations? Please give values p. 3, l. 12-17 This is methodology and should be moved there p. 4, l. 1-7 Idem p. 7, l. 2-20 Idem p. 8, l. 1-4 Idem Figure 2. The text is too small, not readable Response: Thanks for the suggestion. We have corrected all minor suggestions raised. Such as the spacing between 1880 and m followed by "above mean sea level" are corrected. On page 3 of line 3, annual rainfall is added. The interpolated values of

rainfall is also presented on Figure 3. "p. 3, l. 12-17. This is methodology and should be moved there" is suggested. But it is there under Research Methods. The text on Figure 2 modified and can be viewed by zooming the document. Finally, we assure you we conducted detail investigation of the problem and found very useful result that can support land use planners and farmers in the study area. Nonetheless, many of your thoughtful comments and suggestions are considered by the authors and will help us to improve the manuscript further.

Please also note the supplement to this comment:
http://www.solid-earth-discuss.net/se-2016-120/se-2016-120-AC1-supplement.zip

```
┌────────────┐  ┌────────────┐  ┌──────────────┐  ┌──────────┐  ┌──────────┐  ┌────────┐
│ SRTM       │  │ Topo Map   │  │ Meteorological│ │ Geology  │  │2013 SPOT │  │ Field  │
│ 30X30 m    │  │ (1:50,000) │  │ Data         │  │ and Soil │  │ Image    │  │ Data   │
└────────────┘  └────────────┘  └──────────────┘  └──────────┘  └──────────┘  └────────┘
```

Topographic Correction | Contour Digitizing | Transformation (Fourier Index) | | Classification

Re-Interpolate DEM (Spline) 10x10m

Rainfall Map — Soil Texture Map — Landuse Map — Management Practice Map

Flow Accumulation | Flow Length | Slope | R Factor | K Factor | C Factor | P Factor

LS Factor

K*R*LS*C*P

Soil Erosion Risk Map

**Fig. 1.** Figure 2. The conceptual framework of the research methodology

---

## Author Comment (AC2) · 20 Sep 2016

Authors' Response to the Referee #2 Comments Title of Paper: Estimating soil erosion risk and evaluating erosion control measures for soil conservation planning at Koga Watershed, Ethiopian Highlands Authors: Tegegne Molla and Biniam Sisheber We greatly appreciate the time and efforts by Pacheco in reviewing this manuscript and the constructive comments offered. We have included the concepts as suggested.

Comment 1: OUTLINE AND GENERAL APPRECIATION: This is a conventional study on soil erosion rates estimated by the RUSLE equation, with indication of conservation

measures for soil loss attenuation. It brings nothing conceptually new but it provides insights on soil erosion of a specific region of Ethiopia (the Koga watershed). The study is well written and documented and merits publication in Solid Earth, with a minor revision. Response: Thanks for the appreciation and we fully agree with your idea. As you mentioned our purpose is to illustrate scientifically the rate and risk of soil loss by adopting RUSLE to the local context. In this study, we assume that environmental planners can get information about the magnitude of soil erosion and status of SWC structures in the Koga watershed which is the source Blue Nile River and Koga Irrigation Dam.

Comment 2: MINOR COMMENT 1) In lines 15-16 of page 1, the authors should include the environmental land use conflicts (ELUC) as another major cause of soil loss amplification, as recently recognized by Pacheco et al. (2014) and Valle Junior et al. (2014). The ELUC are related to land uses not conforming to soil's capability, meaning that are uses which deviate from the soil's natural use (e.g. practice of agriculture in soils solely capable of being used for forestry). Apart from the amplification of soil erosion, the ELUC have been demonstrated to provoke a decline in soil fertility (Valera et al., 2016). Somehow, these aspects of soil erosion / decline of soil fertility should be referred to in the revised manuscript. Response: Based on the suggestion, we have included the ELUC as major cause of soil loss. The Authors' also added the land use conflict of Koga watershed under the Result and Discussion section. The referred documents are put in the revised manuscript on page 1 as follows: "Soil erosion is further aggravated by environmental land use conflicts (ELUC), as recently recognized by Pacheco et al. (2014) and Valle Junior et al. (2014). The ELUC are uses of the land that ignore soil capability to use and treat according to the soil's natural use. Aggravation of the land use conflicts have been investigated in developing countries which provoke a decline in soil fertility (Valera et al., 2016)."

The citations referred above are written under the reference section as follows: Pacheco, F.A.L., Varandas, S.G.P., Sanches Fernandes, L.F., and Valle Junior, R.F.:

Soil losses in rural watersheds with environmental land use conflicts. Science of the Total Environment, v. 485–486C, p. 110–120, 2014. Valera, C.A., Valle Junior, R.F., Varandas, S.G.P., Sanches Fernandes, L.F., Pacheco, F.A.L. The role of environmental land use conflicts in soil fertility: A study on the Uberaba River basin, Brazil. Science of the Total Environment, v. 562, p. 463–473, 2016. Valle Junior, R.F., Varandas, S.G.P., Sanches Fernandes, L.F., & Pacheco, F.A.L.: Environmental land use conflicts: A threat to soil conservation. Land Use Policy, v. 41, p. 172–185, 2014.

---

## Author Comment (AC3) · 25 Sep 2016

Authors' Response to the Referee #3 Comments Comment 1: This paper shows an interesting topic which is well presented in the Introduction. I would include in line 22 two new appointments: Asensio, C.; Lozano, F.J.; Ortega, E.; Kikvidze, Z. Study on the effectiveness of an agricultural technique based on aeolian deposition, in a semiarid environment. Environmental Engineering and Management Journal, 14 (5): 1143-1150. 2015. Lozano, F. J.; Soriano, M.; Martínez, S.; Asensio, C. The influence of blowing soil trapped by shrubs on fertility in Tabernas district (SE Spain). Land

Degradation & Development, 24 (6): 575-581. 2013.

Response: Thank you for your genuine and constructive comments! The two references you recommended us to include provides very important analysis about the effect of wind on soil erosion. Off course it is one of the factor causing soil degradation in our study area too. However, we didn't discussed about it because of two reasons. Firstly, soil degradation due to blowing wind in Ethiopian highland is very small and is not recognizable compared to huge amount of soil lost every year due to running water. Secondly, the model we chose to (RUSLE) can't take wind erosion in to account. Therefore, the introduction only focused on water erosion. Comment 2: In the section on Material and Methods, I think part of the Description of the Study Site could be extended by reference to characteristics of the site that appear later in the Results and Discussion section. The part of Research Methods, should also be extended with a large number of descriptions in Results and Discussion. In the discussion could include more references to the effect of erosion, in that specific region of Ethiopia, on soil fertility. Response: This is really a critical comment and we appreciate the recommendation. According to your comment, brief explanations about the practice and method of agriculture, land use and the management practice and soil type, which all are causes of erosion are incorporated in each subsection under Results and Discussion section. Moreover, we have supported the analysis and discussion section with additional similar studies in part of the Ethiopian highlands as it further improves this manuscript. However, extending the description of Research Methods might create redundancy as it is depicted on flow chart of the study. The Result and Discussion part also includes the key Research Methods how factors of RUSLE are determined and analyzed.

---

## Referee Comment (RC4) · Anonymous Referee #3 · 27 Sep 2016

Dear authors, obviously, it seems odd jobs include wind erosion references in an article that has RUSLE as a base. However, both recommended references show the protective effect of shrubs in front of the detachment, transport and deposition of soil particles elsewhere, whether those actions are due to wind or water. Besides the role of fertility islands formed underneath the shrubs canopies from semiarid environments, on runoff processes. Anyway, this is just an opinion.

---

## Editor Comment (EC1) · A Jordán (Editor) · 29 Sep 2016

**ABSTRACT**

The abstract is not a simplified version of the manuscript. Use it to engage a wide audience using simple text and avoiding jargon as much as possible. Try to follow this sequence:

- 1. Include a pair of sentences to explain why your research is important and your main objectives.
- 2. Describe methods used, just the general outlines, not in detail.
- 3. Explain your main results and conclusions. More than numbers, you must explain trends and their relevance.

What does "35.56% of soil conservation practices" mean?

**INTRODUCTION**

Only one sentence is devoted to objectives (page 2, lines 22-24 in the online pdf). Rewrite it describing your main/general objective in first place (estimating soil erosion risk and evaluating erosion control measures for soil conservation planning at Koga watershed). Then, your secondary objectives (modelling soil erosion with RUSLE, assessment of SWC measures...).

**RESULTS AND DISCUSSION**

You have chosen to combine both sections in one. I prefer using different sections for each, but in this case, always, make clear what is "results" and what is "discussion". For example: section 3.1 (Rainfall erosivity) starts with a short discussion and results start several lines below. Rearrange each section so that results are shown in the first place.

**FIGURES AND TABLES**

All figures and tables must be completely understood when read separately from text. So, add all details to the captions.

Do not use colored boxes for legends and scale bars (eg, pale yellow for legend and cyan for scalebar in Fig. 5).

Use always the same symbol for the north arrow.

Use always the same scale (and scalebar). In the scale bar, use round numbers for breaks. Try to avoid this:

Instead, for example, use 0, 5 and 10 km breaks and always the same style.

**FIGURE 1**

Coordinates in the frame and in the figure (crosses) are not necessary. Instead: add the name of Ethiopia and reorder the elements, so that the scale bar is undoubtedly visually associated to the studied catchment.

**FIGURE 3**

Rewrite the caption: Map of R factor (left) and rainfall (right). May be obvious to you, but explain clearly that it is mean annual rainfall.